# The Role of WAVE2 Signaling in Cancer

**DOI:** 10.3390/biomedicines9091217

**Published:** 2021-09-14

**Authors:** Priyanka Shailendra Rana, Akram Alkrekshi, Wei Wang, Vesna Markovic, Khalid Sossey-Alaoui

**Affiliations:** 1Department of Medicine, Case Western Reserve University, Cleveland, OH 44106, USA; pxr240@case.edu (P.S.R.); axa1061@case.edu (A.A.); wxw363@case.edu (W.W.); 2MetroHealth Medical Center, Cleveland, OH 44109, USA; vxm196@case.edu; 3Case Comprehensive Cancer Center, Cleveland, OH 44109, USA

**Keywords:** WAVE2, cancer, signaling, phosphorylation, WASP, Arp2/3, Rac1

## Abstract

The Wiskott–Aldrich syndrome protein (WASP) and WASP family verprolin-homologous protein (WAVE)—WAVE1, WAVE2 and WAVE3 regulate rapid reorganization of cortical actin filaments and have been shown to form a key link between small GTPases and the actin cytoskeleton. Upon receiving upstream signals from Rho-family GTPases, the WASP and WAVE family proteins play a significant role in polymerization of actin cytoskeleton through activation of actin-related protein 2/3 complex (Arp2/3). The Arp2/3 complex, once activated, forms actin-based membrane protrusions essential for cell migration and cancer cell invasion. Thus, by activation of Arp2/3 complex, the WAVE and WASP family proteins, as part of the WAVE regulatory complex (WRC), have been shown to play a critical role in cancer cell invasion and metastasis, drawing significant research interest over recent years. Several studies have highlighted the potential for targeting the genes encoding either part of or a complete protein from the WASP/WAVE family as therapeutic strategies for preventing the invasion and metastasis of cancer cells. WAVE2 is well documented to be associated with the pathogenesis of several human cancers, including lung, liver, pancreatic, prostate, colorectal and breast cancer, as well as other hematologic malignancies. This review focuses mainly on the role of WAVE2 in the development, invasion and metastasis of different types of cancer. This review also summarizes the molecular mechanisms that regulate the activity of WAVE2, as well as those oncogenic pathways that are regulated by WAVE2 to promote the cancer phenotype. Finally, we discuss potential therapeutic strategies that target WAVE2 or the WAVE regulatory complex, aimed at preventing or inhibiting cancer invasion and metastasis.

## 1. Introduction

Almost all somatic cells born out of mitosis eventually die by apoptosis—a physiological process by which cells culminate into a programmed cell death or cellular suicide [1,2]. However, when this natural balance between cell growth and death is disturbed, either by increased cell proliferation or decrease in apoptosis, it gives rise to cancer. Invasion is an early step in cancer progression which refers to the ability of cells to become mobile and navigate through the extracellular matrix within a tissue or to travel and infiltrate the adjacent tissues [3]. Metastasis is a complex process by which cancer cells migrate through the blood and the lymphatic systems and proliferate in distant sites and organs in the body. The invasion-metastasis process involves many cellular interactions and signaling pathways [4,5,6,7]. 

Plasma membrane protrusions in the form of filopodia and lamellipodia initiate cell motility and migration [8], which are required for cancer cell invasion of the tumor extracellular matrix [9,10]. Lamellipodia are composed of a dense actin network—a major component of actin cytoskeleton which determines and maintains cell shape. Actin polymerization and depolymerization can cause dramatic changes in cell shape and provide them with migratory properties. The actin-related protein 2 and 3 complex (Arp2/3 complex) present at the corners of branched actin filaments is critical for lamellipodia formation [11,12,13]. The Arp2/3 complex stimulates actin polymerization by creating a nucleation core by ATP hydrolysis, which is an initial step in the formation of actin filaments [14,15,16]. This nucleation core activity of the Arp2/3 complex is activated by its binding to the Wiskott–Aldrich syndrome protein (WASP) family (Figure 1) [17,18]. The WASP family proteins include neural WASP (N-WASP) and WASP family verprolin-homologous protein (WAVE1, 2 and 3). The WASP and WAVE family proteins are analogues and share a common C-terminal verprolin-homology domain (V, also known as WASP homology 2 (WH2)) [19], the cofilin domain (C, also known as central domain) and the acidic domain (A), which collectively form the VCA region [20]. The VCA region activates Arp2/3 complex by binding to actin monomer, leading to actin polymerization [21,22]. Besides the VCA region, the WASP subfamily proteins are conserved in their amino-terminal WASP homology 1 (WH1) domain that allows protein-protein interaction [23]. In contrast, the WAVE subfamily proteins are characterized by the presence of WAVE homology domain (WHD), also known as the SCAR homology domain (SHD) [23,24,25,26]. This domain is located at the amino-terminus and is involved in the formation of WAVE regulatory complex (WRC) [23,24,25,26]. WASP is expressed exclusively in hematopoietic cells [13], while the N-WASP is expressed ubiquitously [27]. In mammals, WAVE2 is expressed ubiquitously whereas WAVE1 and WAVE3 are enriched in the brain but are also expressed throughout the mammalian body [23].

The discovery of WASP protein evolved from the study of Wiskott–Aldrich syndrome; it is an X-linked autosomal recessive disorder due to mutation in the WAS gene and is characterized by recurrent infection due to immune deficiency, microthrombocytopenia and eczema [28]. WASP is involved in transmitting signals from the surface of blood cells to the actin cytoskeleton. These signals allow cells to move and attach to other cells and tissues. Loss of WASP signaling disrupts the function of the actin cytoskeleton in developing blood cells which decreases their ability to respond to their environment and to mount an immune response against foreign invaders, resulting in immune disorders related to Wiskott–Aldrich syndrome. Microthrombocytopenia occurs due to the lack of WASP in platelets which impairs their development, leading to reduced size and early cell death [29,30,31,32,33]. Additionally, platelets are cleared from the circulation by the spleen and bone marrow at a faster rate and this could be due to abnormal shape or structure, or the presence of platelet-associated immunoglobulins [28,34,35,36].

WAVE1, 2 and 3, encoded by WASF1, WASF2 and WASF3, respectively, play a critical role downstream of Rac, a Rho family small GTPase. They are involved in the formation of the WAVE regulatory complex (WRC) which in turn regulates the actin cytoskeleton [37]. The WRC is also associated with actin nucleation core Arp2/3 complex while promoting actin polymerization at the leading edge of lamellipodia [38]. WAVE family proteins interact with brain-specific angiogenesis inhibitor 1-associated protein 2, also known as IRSp53 [39,40]. According to published reports, the WAVE1 expression becomes increasingly restricted to CNS over the course of development, and homozygous disruption of WAVE1 gene has been shown to result in postnatal lethality [41]. Several other developmental disorders include severe limb weakness, tremors, neuroanatomical malformations, etc. [41]. Other studies have shown that WAVE2-deficient embryos display growth retardation and some morphological defects such as malformation of ventricles in the developing brain. The WAVE2-deficient embryonic fibroblasts were shown to exhibit severe growth defects [42]. Meanwhile, no abnormal development or functions of mouse mammary gland or brain development have been observed in the absence of WAVE3 [43]. Since WASP and WAVE proteins are required for several biological processes, such as formation of filopodia and lamellipodia, and provide cells with the migratory property, they have recently been a topic of great interest in cancer invasion and metastasis. The main focus of this review revolves around WAVE2 in its potential role in the pathogenesis of different types of cancers and its signaling events that lead to cancer development, invasion and metastasis.

## 2. The WAVE Regulatory Complex

The WAVE regulatory complex (WRC) is a stable pentamer of WAVE proteins; Abelson interacting protein (ABI); NCK-associated proteins (NAP), specifically Rac-associated 1 protein (SRA1); and hematopoietic stem progenitor cell 300 (HSPC300) [44,45,46]. The WRC activates Arp2/3 complex which then causes actin polymerization which in turn induces protrusion events necessary for cell movement during development, new axon and dendrite formation during neural development wound healing, immune responses and cancer cell metastasis [47].

The WRC is a very stable complex and knockdown or knockout of any one of its components decreases the levels of the remaining components, which makes it an intermolecularly inhibited complex. The WAVE VCA region stimulates actin nucleation [18,48,49] by upstream signals needed to activate the WRC. These signals cause conformational change that frees the VCA region to bind to Arp2/3 and actin (Figure 2).

The WRC is also a signal integrator. In response to extracellular stimuli, signals form Rac GTPase, and phospholipids and kinases come together upon components of WRC to induce its localization and activation. Rac-GTPase binds to the WRC through SRA1 [50]. This interaction competes with sequestration of WAVE VCA region and makes it accessible to Arp2/3 complex and actin.

Binding of negatively charged phospholipids recruits the WRC complex to cell membrane which leads to the activation of Arp2/3 complex. For e.g., phosphatidylinositol-(3,4,5)-triphosphate (PtdIns(3,4,5)P3) and phosphatidylinositol-(4,5)-diphosphate (PtdIns(4,5)P2) bind to WAVE2 [51,52]. This lipid binding is essential for translocation of WAVE2 to the leading edge and Rac-induced protrusion [52]. There also have been reports of other negatively charged lipids inducing partial WRC activity in a charge-dependent manner [38].

WRC integrates intracellular and extracellular cues via multiple signaling pathways. Cytokines, hormones, growth factors, neurotransmitters and extracellular matrix signals activate the following kinases—Abl, Src and ERK, which in turn promote activation of WRC. Phosphorylation by Abl, Src and ERK regulate WRC localization to cell membrane and function to control the actin cytoskeleton. Whereas the neuron-specific CDK5 and constitutively active CK2 kinases inhibit the activation of WRC (Figure 3).

## 3. Role of WAVE2 in Cancer

The first reports linking any WASP or WAVE protein to cancer pathology date back to 2005 when the Takenawa group, which was also heavily involved in the identification and characterization of the WASP/WAVE gene family [53,54], first described a mouse model where WAVE2 plays a major role in driving the invasive and metastatic phenotypes of murine melanoma [55]. The second study was reported by the Sossey-Alaoui group; that was the first study to describe the role of WAVE3 in promoting breast cell motility and invasion [56]. These pioneering works set the stage for several studies from different research groups that investigated and established members of the WASP/WAVE gene family [57], including WAVE2 as a driving force behind the aggressiveness of several human malignancies, both of solid and hematological origin. The sections below will review the most relevant literature related to the role of WAVE2 in these malignancies (Figure 2). 

### 3.1. WAVE2 in Colorectal Cancer

Colorectal cancer is the third most common cause of cancer-related deaths in both men and women in the United States and worldwide. It has accounted for about 7.88 percent of 2021 cancer diagnoses in both men and women and 8.7 percent of all cancer deaths in the US in the year 2021 [58]. There are several reports that show expression of WAVE2 in colorectal cancer tissues as well as its association with liver metastasis, disease progression and microvessel density and its correlation with activation of the TGF-β1 and YAP1 signaling pathways. It was found that TGF-β1 is involved in the cancer immune microenvironment and promotes metastasis of liver tumors [59]. Studies have also shown that TGF-β1 mediates WAVE2 upregulation on hepatic stellate cell (HSC) activation and promotes activated-HSCs metastasis. Thus, these findings imply that WAVE2 plays a critical role in colorectal liver metastasis [60,61,62,63]. 

### 3.2. WAVE2 in Cervical Cancer

According to most recent cancer statistics, cervical cancer is the fourth most common cancer in the United States [58]. It has accounted for about 1.5 percent of 2021 cancer diagnoses in women and 1.48 percent of all cancer deaths (in women) in the year 2021 [58]. According to the American Cancer Society, cervical cancer is most commonly diagnosed in women between the age group of 35–44 years [58].

Overexpression of SH3-domain-binding protein-1 (SH3BP1) promotes invasion, migration and chemoresistance of cervical cancer cells through increasing the activity of Rac1 and WAVE2. Moreover, it has been shown that in the cisplatin-resistant cervical cancer tumors, expression of SH3BP1, Rac1 and WAVE2 mRNA is significantly upregulated compared to cisplatin-sensitive cancer tissues [64].

### 3.3. WAVE2 in Pancreatic Cancer

Pancreatic cancer is the fourth leading cause of cancer-related fatalities in the United States in the year 2021 [58], with most patients being asymptomatic until the disease reaches an advanced stage [65]. According to recent statistics, pancreatic cancer accounts for about 3% of all cancers in the US and about 8% of all cancer deaths [58]. Taniuchi et al. [65] reported the role of WAVE2 in motility and invasiveness of pancreatic cancer cells [65]. They showed how accumulation of WAVE2 increased cell protrusions, causing invasiveness and motility of pancreatic cell lines, and how the downregulation of WAVE2 had an opposite effect [65]. They also reported that pancreatic cells become motile and invasive by forming a complex with actin cytoskeletal protein alpha-actinin 4 (ACTN4), downregulation of which decreases cell protrusions [65]. Their results suggest that the signaling of WAVE2/ACTN4 stimulates p27 phosphorylation which imparts motility and invasiveness to these cells [65]. 

In another study, they have also reported that the use of siRNA against WAVE2 resulted in inhibition in invasiveness and metastasis in pancreatic ductal adenocarcinoma cells by decreasing the cell protrusions [66].

Kitagawa et al. showed that the serum level of WAVE2 mRNA was most highly correlated with the risk of PDAC. They concluded that WAVE2 may be a noninvasive diagnostic biomarker for the early detection of PDAC [67].

### 3.4. WAVE2 in Prostate Cancer

According to 2021 statistics, prostate cancer is the first leading cancer among men and is the second leading cause of cancer death, behind only lung cancer in the United States [58]. In the United States, about 1 in 8 men will be diagnosed with prostate cancer during his lifetime. Prostate cancer has accounted for about 13.1% of new cases in men and 10.7% of all cancer deaths in 2021 [58].

Since WAVE2 plays a major role in Rac1-induced actin reorganization in association of PIP3, there have been reports suggesting its involvement in prostate cancer cells. Kato et al. reported the role of WAVE2 with PIP3 in lamellipodial extension which underlies prostate cancer cell invasion and metastasis [68].

### 3.5. WAVE2 in Breast Cancer

According to WHO, breast cancer became the most common cancer in women globally as of 2021, accounting for 12% of all new annual cancer cases worldwide [58]. About 1 in 8 U.S. women (about 13%) will develop invasive breast cancer over the course of their lifetimes. Breast cancer is the most frequently diagnosed malignancy in women and is one of the leading causes of death due to cancer invasion, metastasis and resistance to therapies [69]. Among its variants, triple-negative breast cancer (TNBC) is considered the most aggressive due to its early invasive and metastatic properties with poor prognosis. TNBC is a molecular and clinicopathological subtype of breast cancer that lacks expression of the receptors that are commonly found in this type of cancer (estrogen, progesterone and HER2), which makes it a more difficult target for treatment [70]. Many studies have reported that the formation of excessive protrusions in breast cancer cells is a result of Arp2/3 complex, which initiates the actin reorganization to form lamellipodial protrusions by binding to WAVE2 [71,72]. Thus, WAVE2 and Arp2 coexpression is a significant prognostic factor which is closely associated with aggressive morphology of breast cancer.

In one study, Hyun Suk Ko demonstrated that inhibiting the Rac1/WAVE2/Arp2/3 pathway suppressed the migration and invasion of breast cancer MDA-MB-231 cells without any cytotoxicity [73]. Additionally, Nicole S. Bryce reported that the WAVE2 complex plays an important and complex role in 3D morphogenesis of breast epithelial cells. They also identified a partial EMT-like phenotype in WAVE2 knockdown cells, with elevation in TWIST levels and cadherin switching. Overall, their studies present an intricate role of WAVE2 complex in controlling breast epithelial morphogenesis through Abl kinase and the transcription factor Twist1 [74]. Kazuhide Takahashi reported that depletion of WAVE2 by RNA interference abrogated both cell invasion and intensive F-actin accumulation at the invasion site, which indicates that WAVE2 is one of the crucial components for PI3K-dependent cell invasion induced by PDGF [75].

It has also been reported that the binding of WAVE2 to Arp2/3 complex is the final signal for triggering the formation of lamellipodia that initiate directional migration of mammalian cells. This signal is enhanced in some breast cancer cell lines such as MDA-MB-231, SKBR3, AU565 and MCF7. Previous studies have shown that the expression of WAVE2 and Arp2 significantly correlated with the overexpression of HER2 (a promoter of cancer cells). This indicates that the signal from HER2 facilitates binding of WAVE2 to activated Rac1 via one of the products of PI3K, phosphatidylinositol-(3,4,5)-triphosphate in HER2-overexpressing cells, which in turn enhances the signal for lamellipodium formation inducing abnormal cell movement [76].

## 4. WAVE2 Signaling Cascade

To study the exact contributions of WAVE2 towards stages of cancer progression, it is important to determine its molecular mechanism and signaling cascade. WAVE2, amongst the WAVE family proteins, has a central role as a switch that initiates polymerization of actin for the formation of lamellipodia and initiation of the amoeboid movement [77,78]. The interaction between the Arp2/3 complex and WAVE2 occurs at the leading edge downstream of the signaling pathway which is responsible for the directional movements in response to several different stimuli—epidermal growth factor (EGF) and platelet-driven growth factor (PDGF) [12,23]. Phosphatidylinositol-(3,4,5)-triphosphate (PIP3) is generated upon activation of these receptors, which in turn activates Rac on the cell membrane. WAVE2 can regulate activation of Arp2/3 complex from the signals of PIP3 binding and activated Rac [52].

The majority of intracellular WAVE2 is complexed with Abl-interactor-1 (Abi1), Nck-associated protein-1 (Nap1), a small protein HSPC300 and p53-inducible protein-121 (PIR121)/Sra1 [79]. So far, it is known that the activity of WAVE2 is regulated by other molecules that act as different signaling inputs regulating the dynamics of the actin cytoskeleton.

## 5. Activation of WAVE Regulatory Complex 

WAVE2 is activated by binding to Rac1, in response to extracellular stimuli, via a large WAVE2 protein complex that includes Sra1, Nap1, Abi1 and HSPC300 or by Rac1-IRSp53 signaling pathway. Amongst these, the IRSp53, an insulin receptor substrate with an unknown function is associated with WAVE most strongly [80]. The IRSp53 is known as a linker molecule that connects Rac1 and WAVE2 [40]. In comparison, WAVE2 has much stronger affinity for IRSp53 than WAVE1 or WAVE3, which suggests that the WAVE2-IRSp53 interaction primarily contributes towards the regulation of WAVE2 [40]. There is additional evidence that the activated Rac binds to the Rac-binding domain (RCB) which is also known as the inverse BAR (I-BAR) domain [81] in the N-terminus of IRSp53. Whereas the C-terminally located SH3 domain of IRSp53 binds to the proline-rich sequences of WAVE2, in turn enhancing the activity of WAVE2 [23]. Moreover, the IRSp53 recruits the WAVE2 complex to the plasma membrane because the RCB of IRSp53 interacts electrostatically with the negatively charged phosphoinositides PI(4,5)P2 and PI(3,4,5)P3 [39,40,82]. This causes the VCA region of WAVE2 to be exposed (which is initially inhibited by its intracomplex interaction with Sra1-NckAP1 subcomplex) [38,83]. The binding of activated Rac to Sra1 triggers a conformational change in the VCA region, causing it to be released from the complex [38,83]. According to the previous studies, there is evidence that the WAVE2 complex is activated in the presence of Rac and negatively charged phospholipids. Binding of PI(3,4,5)P3 to the basic amino acid cluster found in WAVE2 protein activates the complex by causing the recruitment of WAVE2 to the plasma membrane [52]. The exposure of VCA region activates the Arp2/3 complex which in turn leads to polymerization of actin filaments and the formation of lamellipodia (Figure 1). Thus, the WAVE2-Arp2/3 forms lamellipodium by nucleation of actin assembly. 

The inactive WAVE2 forms a complex with the IQ motif containing GTPase activation protein 1(IQGAP1) and kinesin family member 5B (KIF5B), which is a component of a motor protein complex that regulates and mediates the transport of cytoplasmic granules and molecules of RNA in migrating breast cancer cells [84]. It has also been shown that when WAVE2 and KIF5B are transported to the cell protrusions, the frequency of cells with lamellipodia increases [65].

Overall, the signals responsible for increasing the cellular levels of PI(3,4,5)P3 are elevated by mutations in genes that constitute the PI3K pathway by either activation mutations or loss of function mutations—for e.g., activating mutations in PI3KCA and loss of function mutations in PTEN [85]. In addition to the WAVEs, the PI(3,4,5)P3 also activates Rac through the guanine-nucleotide exchange factors for Rac (RacGEFs) [86,87]. Therefore, the WAVE complex is known to be hyperactivated in cancers which have elevated PI3K signaling. Additionally, WAVE has been reported to be regulated by phosphorylation events such as c-Abl tyrosine kinase activating WAVE2 through phosphorylation [88].

## 6. WAVE2 Is Regulated by Several Phosphorylation Events

Extracellular signals such as growth factors, cytokines and adhesion status can be transmitted to the cell machinery by the phosphorylation mechanism (Figure 3). Kinases are the enzymes involved in the transfer of the (gamma) phosphoryl group (PO_3_^2−^) from an ATP molecule to serine, tyrosine or threonine residues in the WAVE regulatory complex [47]. Phosphorylation of WAVE2 constitutes a form of regulation that influences fundamental aspects of cell motility [47].

### 6.1. AbI Tyrosine Phosphorylation of WRC

The non-receptor tyrosine kinase AbI phosphorylates tyrosine residues, and its activity is in turn regulated by cytokines, growth factors and cellular differences in its nuclear vs. cytoplasmic localization [89]. According to two reports [90,91], phosphorylation of Abl and activation of WRC partly mediate cytoskeletal remodeling, cell migration and leukemia. The SH3 domain and the proline-rich sequences on the Abl are required for its interaction with the other WRC members and for ABI phosphorylation of WAVE2 [88,92,93,94]. AbI phosphorylates WAVE2 on its conserved Tyr 150 [88,93]. The phosphorylation reaction disturbs the interaction between the WAVE meander region and SRA1 which exposes the VCA region to reaction with the Arp2/3 complex (Figure 2) [49]. Thus, AbI phosphorylation of WAVE is necessary for WRC-mediated activation of Arp2/3 complex and actin polymerization.

### 6.2. SRC Tyrosine Phosphorylation of WRC

The Src non-receptor tyrosine kinase phosphorylates Tyr121 of WAVE2 which is also conserved in WAVE1 and WAVE3 [46,95]. Like Tyr150, the Tyr125 phosphorylation also destabilizes the meander region of WAVE which releases the VCA region to interact with the Arp2/3 (Figure 2) [46]. Thus, Src promotes actin polymerization through phosphorylation of WAVE2 [96,97,98].

### 6.3. Serine/Threonine Phosphorylation of WRC

#### 6.3.1. Cyclin-Dependent Kinases (CDKs)

The cyclin-dependent kinases are a family of proline-directed protein kinases that are known for their role in regulating the cell cycle. Amongst these, CDK1 phosphorylates serine 216 on ABI1 [92], which attenuates ABI1-stimulated tyrosine phosphorylation of WAVE2 and causes inhibition of F-actin assembly in the leading edge while oncogenic Abl increases the percentage of cells with F-actin rich spots (Figure 2).

On the other hand, CDK5 has been known to phosphorylate WAVE2 at Ser137 in oligodendrocytes (glial cells of the central nervous system) [99] which are known to inhibit the activity of WAVE2 (Figure 2) [46].

#### 6.3.2. Casein Kinase 2 (CK2)

CK2 is a serine/threonine kinase that is constitutively active and has been implicated in cell cycle regulation, DNA repair and other cellular processes. Improper regulation of CK2 has been linked to cancers of breast, lung, colon and prostate [100]. CK2 phosphorylates the VCA region (acidic domain) of WAVE2 on Ser482, 484, 488, 489 and 497 [101]. Ser489 is unique to WAVE2 and Ser497 is unique to WAVE2 and WAVE3, whereas Ser482, 484 and 488 are conserved in all WAVE1, WAVE2 and WAVE3 [53]. CK2 phosphorylation of these five sites is essential for the interaction with Arp2/3 complex [101]. The acidic domain of VCA region has been proposed to form an intramolecular interaction with the basic region on WAVE that strengthens the CK2 phosphorylation [102]. Phosphorylation of WAVE2 VCA domain is critical for normal WAVE2 function and any mutation in the phosphorylation inhibits WAVE2 function in vivo, inhibiting the cell ruffling and disrupting the lamellipodium formation [101]. Constitutively active CK2 has been known to phosphorylate the VCA domain of WAVE to inhibit the WRC activation (Figure 2). This has been proposed to cause the loss in persistent cell migration observed in WAVE2-depleted cells [103]. 

#### 6.3.3. ERK

ERK is a type of serine/threonine protein kinase (closely related to CDK) that transmits mitogen signals and functions as an effector of Ras oncoprotein. ERK is known to phosphorylate WAVE2 on Ser343, Thr346 and Ser351 within the PRD [103,104,105] and in some cell lines it phosphorylates WAVE2 on Ser308 [104,105]. Mendoza et al. [47,104] showed that ERK phosphorylation of WRC on multiple sites within the PRD of WAVE2 and Abi1 contributes towards Rac-induced WRC conformational change which exposes the VCA domain, leading to WRC activation (Figure 2) and lamellipodial protrusion [104].

#### 6.3.4. Protein Kinase A

PKA is a serine/threonine kinase which is known to be activated through the release of the catalytic subunits when the levels of cAMP (a secondary messenger) rise in response to a myriad of signals. PKA phosphorylates the regulatory subunits of the protein phosphatase 2A (PP2A) which causes the activation of PP2A. Activated PP2A is reported to mediate the dephosphorylation of inhibitory CDK5 sites on WAVE1 [106]. PKA has also been known to physically interact with WAVE1 and WAVE2 to cause increase in the local concentration of PP2A near WAVE2 regulatory complexes [107,108]. Overall, PKA activates phosphatase to remove the inhibitory CDK5 phosphorylation [107,108].

## 7. WAVE2 and MicroRNAs

MicroRNAs are small, non-coding RNA which are about 18–23 nucleotides in length. They are known to regulate gene expression by binding to the 3′ untranslated region (UTR) of a target mRNA. These microRNAs can either trigger translation repression or mRNA cleavage and are thus responsible for RNA silencing and post-transcriptional regulation of gene expression. There are several tumor-suppressive miRNAs in breast cancer—miR-30a, miR-30c, miR-31, miR-126, miR-133b, miR-140, miR-146b, miR-181c, miR-200c, miR-206 and miR-335 [109,110,111,112,113,114,115,116,117].

MiR-181c indirectly suppresses WAVE2 expression levels in cancer cell lines. Suppression of BRK1 as a result of miR-181c overexpression has been known to reduce protein expression of WAVE2 complex (WAVE2, ABI1, SRA1 and HEM1) in HeLa [80], osteosarcoma U2OS cell lines, Jurkat and primary T cells [118,119,120].

The miR-133b is known to regulate tumorigenesis and metastatic potential of breast cancer [121]. Qiu-Yu Wang [121] demonstrated for the first time that miR-133b is pathologically downregulated in breast cancer specimens and cell lines, whereas the ectopic expression of the same was involved in strongly suppressing the metastatic-relevant traits in the human breast cancer cell lines [121]. Collectively, their results suggest that miR-133b, a novel tumor suppressor in breast cancer, could suppress the tumor invasion and metastasis by downregulating its target WAVE2.

## 8. Regulation of WAVE2 by Oncogenic Signals

The activation mechanisms of WAVEs with respect to cancer pathways have been extensively studied in recent years [122,123,124]. In the following section, we discuss how the activity of WAVE2 is affected by several oncogenic signals.

### 8.1. Protein Kinase B (AKT)

Dysregulation of AKT has been proposed to promote proliferation, migration and increased cell survival, thereby contributing towards cancer progression. Increased Akt kinase activity has been reported in various cancers—breast, ovarian, prostate and gastric [125,126].

Moraes, L. et al. [127] showed that AKT interferes in WRC formation by modulation of Abi3 phosphorylation state. According to their reports, phosphorylation of ABI3 by AKT inhibits the formation of an inactive WRC and thereby releases the WRC complex to an open/activated WAVE2 (Figure 3). However, in the presence of AKT inhibitor, the ABi3 remains unphosphorylated in an inactive WRC which confirms that AKT is a positive regulator of WAVE2 [127].

### 8.2. TGF-β

TGF-β, a multifunctional cytokine, has been known to play a paradoxical role in cancer as a tumor suppressor and a tumor promoter [128]. In normal cells, TGF-β arrests the cell cycle at G-1 phase to stop proliferation to promote differentiation and apoptosis. However, in cancer cells, the TGF-β pathway is mutated and can no longer control cell death, which causes the cells to proliferate. This TGF-β acts on the surrounding immune cells to cause immunosuppression and angiogenesis which cause cancer invasion [129]. 

In colon cancer liver metastasis (CLM) patients, WAVE2 expression was strongly correlated to microvessel density in hepatic metastasis. Moreover, WAVE2 promotes the progression of CLM by regulating TGF-β and Hippo pathways via effector yes associated protein (YAP1) in the hepatic stellate cells [62].

### 8.3. VEGF

Vascular endothelial growth factor (VEGF) is a signaling protein which is a potent and specific angiogenic factor [129]. It is known for its ability to induce vascular permeability to stimulate endothelial cellular growth and hence growth of new blood vessels. VEGF is known to induce tumor growth through angiogenesis [130]. According to Tao et al., VEGF secretion is elevated via Rac1-WAVE2 signaling (induced by SH3BP1 which promotes angiogenesis, venous infection and tumor invasion in hepatocellular carcinoma) [131]. They found that SH3BP1, a Rho GTPase activating protein, induces the Rac1-WAVE2 signaling to promote the invasion [131]. In yet another study, the binding of VEGF to VEGFR2 caused autophosphorylation of the receptor, which initiates the recruitment of the SH3BP1 adaptor protein [132]. This recruitment then activates PI3K, which in turn activates Rac1 (Figure 3). Rac1 is known to regulate migration through its effector WAVE2 and profilin (actin-binding protein).

### 8.4. ERK-MAPK

A cell responds to the extracellular and intracellular cues by one of the signaling cascades known as the ERK-MAPK. Defects in the ERK-MAPK pathway lead to uncontrolled cell growth in many cancers. ERK-MAPK signaling with respect to WAVE2 and cancer metastasis still requires more investigation, but one article accounts for involvement of ERK-MAPK with WAVE2 in actin assembly and cell migration [104]. Mendoza et al. [104] reported that ERK colocalizes with ERC at lamellipodial leading edge protrusion and directly phosphorylates two WRC components: WAVE2 and Abi1 (Figure 3). This interaction activates Arp2/3 actin nucleator for actin assembly and polymerization to promote cell migration [104]. Therefore, dysfunction in the ERK-MAPK pathway may lead to WAVE2 upregulation which could, in turn, promote metastasis in several types of cancer.

## 9. Possible Targets for Drug Therapies for Preventing Metastasis and Invasion

In the previous decade, we have seen significant advancement in understanding the role of WASPs and WAVEs in cancer research. However, there are still certain links that are yet to be explored in the WASP and WAVE signaling cascade which could be eventually used as targets for potential anticancer drugs. Rhapontigenin and pterostilbene were recently investigated as inhibitors of breast cancer cell metastasis that may act through the WAVE2 pathway in MDA-MB-231 cells which are highly metastatic [73,133]. It was suggested that this effect was a result of impairment in the Rac1-WAVE2 pathway causing a reduction in Rac1 activity, which in turn decreases the expression of downstream effectors such as WAVE2, Arp2 and Arp3 [73,133]. Moreover, it has also been implied with evidence that treatment with rhapontigenin acts to inhibit invasive and migratory behavior in breast cancer cells through inhibition of formation of lamellipodia [73,133].

Since ROCK-inhibitor treatment shifts single disseminating cancer cells towards WAVE2-dependent migratory mode [134], WAVE2 and its associated molecules, IRSp53, would be an attractive target for anti-invasion drug development. The combined inhibition of ROCK and drugs that inhibit WAVE2 activity would be an effective treatment against broad spectrum cancer cell invasion and metastasis. However, there still needs to be a clear understanding of which contributions of WASPs and WAVE cause cancer invasion and metastasis in order for them to be validated as specific anticancer targets. 

## 10. Discussion

WAVE2 is known to be overexpressed in several types of cancers where it plays a significant role in tumor development, metastasis and invasion, which makes it a potential target for drug therapies. Despite dramatic advancements in our understanding of WAVEs in cancer cells, it still remains unclear as to which molecule in WAVE signaling pathway could be targeted for an anticancer drug therapy. It is known that activation of Rac inhibits Rho-ROCK signaling via WAVE2 and drives cells into mesenchymal migration, similar to the migration of fibroblasts [135]. This WAVE2-dependent motility due to ROCK inhibition serves as an attractive target for anticancer therapies. Currently, no ROCK inhibitor is used for clinical trials for treatment of cancer, but many ROCK inhibitors such as Y27632 [136] and Fasudil [137] have shown significant effects in both cancer cell lines and rodent cancer models, supporting the overall importance of ROCK-WAVE2 signaling in the development and progression of cancer. Moreover, determining what combinations of phosphorylation and other signaling events induce increased activation of WAVE2 could also serve as a significant step towards understanding the potential drug therapies for controlling the metastasis and invasion cascade in cancer. However, to further validate WAVE2 as an anticancer target, it is necessary to determine exactly how the upstream and downstream pathways interplay to drive cells into various modes of cancer invasion. With increasing knowledge of regulation and localization of WAVE2 in different cellular settings, the prospect of producing anticancer drugs targeting WAVE2 will become less complicated. The overall goal is to improve the disease outcome by reducing the tumor growth and metastasis in cancer.

## Figures and Tables

**Figure 1 biomedicines-09-01217-f001:**
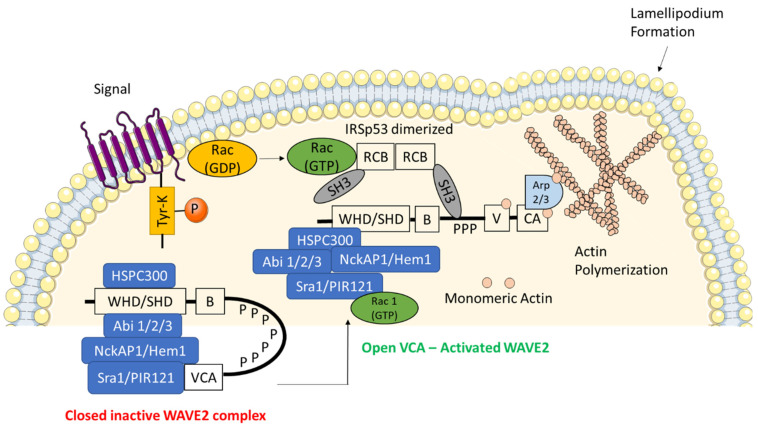
Structural domains and components of active and inactive WAVE regulatory complex (WRC). The multi-protein complex WRC is a stable heterocomplex consisting of HSPC300, wave homology domain (WHD), Abl interactor 1/2/3 (Abi1/2/3), NCK associated protein 1 (NckAP1/Hem1), specifically Rac1-associated protein 1 (Sra1/PIR121). WAVE2 is autoinhibited in a basal state through the interaction between the complex proteins and the VCA region. Upon certain phosphorylation signals the closed complex is released into an open and active WRC which is then translocated to the cell membrane via interactions with the phosphatidylinositol-(3,4,5)-triphosphate (PIP3) and insulin receptor substrate of 53 kDa (IRSp53). The binding of Rho family of GTPases (GTP-Rac) increases the affinity of WAVE2 for IRSp53. This IRSp53 promotes the ability of WAVE2 to stimulate actin-related protein 2/3 complex (Arp2/3)-mediated actin polymerization.

**Figure 2 biomedicines-09-01217-f002:**
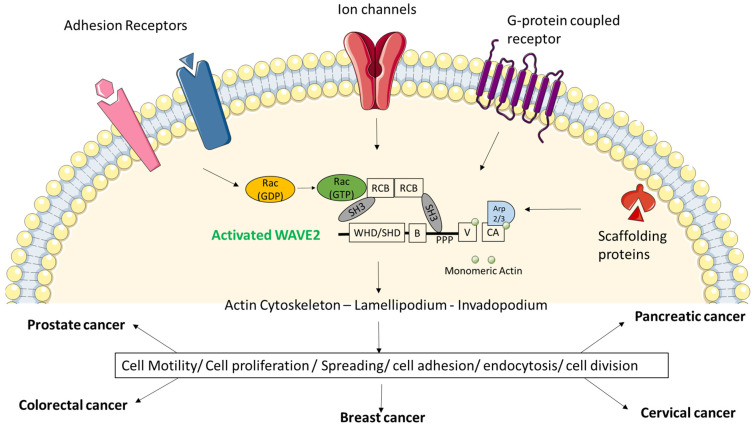
Upstream signals via membrane proteins cause activation of WAVE2 and have broad physiological and pathological ramifications. The potential WRC ligands or membrane proteins such as ion channels, G-protein-coupled receptors, adhesion receptors and scaffolding proteins are involved in recruiting WRC towards the cell membrane which causes polymerization of actin cytoskeleton. This event is extremely crucial for cell structure, migration, spreading, adhesion, division and invasion. High expression of WAVE2 and its constitutively active state is responsible for pathological conditions such as cancer.

**Figure 3 biomedicines-09-01217-f003:**
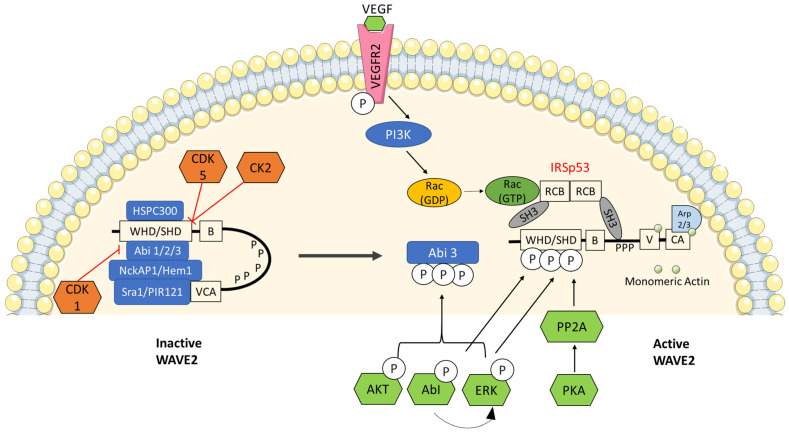
Regulation of WAVE2 by several phosphorylation events: WRC is known as a signal integrator for sensing signals from Rac GTPases, phospholipids and protein kinases that sense growth factors and substrate adhesions. Cyclin-dependent kinase 1—CDK1, CDK5 and casein kinase 2 (CK2) are known to inhibit WAVE2 by keeping it in an inactive state whereas, AKT, ABI, Erk, PKA and PI3K (through VEGF) are known to release inactive WRC into an active WAVE2 complex.

## Data Availability

Not Applicable.

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
