# Peer review of "The Role of WAVE2 Signaling in Cancer"

_biomedicines, 2021, doi:10.3390/biomedicines9091217_

Round 1
Reviewer 1 Report
Comments for Rana et al
The paper appears to be an excellent summary of WAVE2 and cancer. However, I would like to point several things to improve the manuscript.
Figure 1 has the actin filaments penetrating the membrane. Therefore, the membrane should be drawn in a way that wraps the filaments.
The illustrations of WRC should have the interaction with Rac1 at Sra1.
RCB is also called the I-BAR domain and the membrane binding sites. This fact should be cited.
IRSp53 can also bind to Cdc42 through the partial CRIB motif, which should be cited and maybe be discussed.
Author Response
Subject: Revision and resubmission of manuscript number: and response to the reviewers
Our Response
Dear Reviewer,
We would like to thank you for extending us the opportunity to submit a revised draft “ROLE WAVE2 SIGNALING IN CANCER” for publication in Biomedicines Journal. We appreciate your time and efforts to review and improve the article and providing the necessary feedback. We have incorporated most of the suggestions made by the reviewers. These changes are marked in red text within the manuscript. Please see below for a point-by-point response to the reviewer.
Reviewer 1:
Comments and Suggestions for Authors
Comments for Rana et al
The paper appears to be an excellent summary of WAVE2 and cancer. However, I would like to point several things to improve the manuscript.
- Figure 1 has the actin filaments penetrating the membrane. Therefore, the membrane should be drawn in a way that wraps the filaments. –
Our Response: Thank you for this remark. We agree with the reviewer. In figure 1, the membrane has been redrawn in a way that it wraps around the actin filaments
- The illustrations of WRC should have the interaction with Rac1 at Sra1.
Our Response: We agree with the reviewer’s suggestion and the Sra1 and Rac1 interaction has been added to the figure 1
- RCB is also called the I-BAR domain and the membrane binding sites. This fact should be cited. –
Our Response: We agree with this suggestion and the revised text now reads as “There is additional evidence that the activated Rac binds to the Rac-binding domain (RCB) which is also known as the Inverse BAR (I-BAR) domain [81] in the N-terminus of IRSp53”
Reference 81 has been added to support the statement.
- IRSp53 can also bind to Cdc42 through the partial CRIB motif, which should be cited and maybe be discussed.
Our Response: Thank you for this noticing this but according to some authors, the N-WASP causes the formation of filopodia downstream of Cdc42 and WAVE2 causes the formation of lamellipodia downstream of Rac. The main intension was to discuss the primary signal that drives that drives WAVE2 regulatory complex into an active state which happens through Rac binding to IRSp53.
Please find reference 112 to support this notion.

Reviewer 2 Report
ROLE WAVE2 SIGNALIN IN CANCER
Shailendra Rana et al., review submitted to Biomedicines 2021- August 2021
In this review, the authors explore the roles of WAVE2 during cancer progression and potential therapeutic strategies. This review raises interesting concepts and will be of interest to the biomedicines readers once major and minor concerns are addressed. Overall, we suggest that the authors edit their manuscript to prevent duplication of messages to improve the quality of the review (see examples below). The authors need to significantly increase the number of references throughout their manuscript since several statements are made that lack appropriate references. Moreover, we have listed examples of minor points that prevent publication of this manuscript in its current state, but this should be addressed beyond these examples.
Major comment :
- Edit the text to prevent duplication of message. For example, the author often refers to WAVE2 regulating invasion and lamellipodial protrusions, such as line 258, line 272, line 276 This dilutes the message and prevents depth in the review. Line 461 and line 482 have the same message.
- Significant improvements must be made to the reference sections. A large fraction of scientific statements is made without listing the appropriate references.
- We suggest discussing the molecular basis of the WAVE complex before listing the different type of cancers. Section 3 before section 2.
Minor comment :
- Verify postal code in affiliation.
- Line 17 and 18 are redundant in the abstract.
- Line 37. Is invasion really the initial step in cancer progression? Re-word
- Line 46, there seems to be an additional space between shape and Actin.
- Line 49, a space is missing between formation and the bracket of the reference.
- Line 54, a space is missing between the comma and 2.
- Line 57, which collectively form a or the VCA region.
- Line 71, add references regarding the Wiskott-Aldrich syndrome and mention in one sentence what is the syndrome about.
- Line 72 is redundant since the expression pattern of WASP/WAVE is already mentioned line 57 to 61.
- Line 78 add references for the role of WASP in blood and immune response.
- Line 80 add references
- Line 85 add references etc…
- Line 102, downstream of the small Rho GTPase Rac
- Line 105 reference
- Line 109, complex, which
- Line 158, that the cisplatin-resistant, remove the “in”
- Line 183, according to stats, re-word this sentence.
- Line 196, TNBC is a clinical subtype, not a type of breast cancer.
- Line 209, complex role of WAVE2 complex, reword to remove duplication.
- Line 284, format of reference is not the same
Figure comments:
- Figure 1. The lamellipodium formation seems to be outside of the cell membrane. Modify the cell membrane do that it reflects that the actine polymerization in inside the cell. Also, the consequences of activation on the other component of the WAVE2 complex when they are in the open conformation? Do they maintain their interaction, they should be represented on the figure when activated?
Author Response
Subject: Revision and resubmission of manuscript number: and response to the reviewers
Dear Reviewer,
We would like to thank you for extending us the opportunity to submit a revised draft “ROLE WAVE2 SIGNALING IN CANCER” for publication in Biomedicines Journal. We appreciate your time and efforts to review and improve the article and providing the necessary feedback. We have incorporated most of the suggestions made by the reviewers. These changes are marked in red text within the manuscript. Please see below for a point-by-point response to the reviewer.
Reviewer 2:
Comments and Suggestions for Authors
ROLE WAVE2 SIGNALIN IN CANCER
Priyanka Rana et al., review submitted to Biomedicines 2021- August 2021
In this review, the authors explore the roles of WAVE2 during cancer progression and potential therapeutic strategies. This review raises interesting concepts and will be of interest to the biomedicines readers once major and minor concerns are addressed. Overall, we suggest that the authors edit their manuscript to prevent duplication of messages to improve the quality of the review (see examples below). The authors need to significantly increase the number of references throughout their manuscript since several statements are made that lack appropriate references. Moreover, we have listed examples of minor points that prevent publication of this manuscript in its current state, but this should be addressed beyond these examples.
Our Response: Thank you for suggesting this. We agree with the reviewer’s suggestions. The appropriate references are added beyond the reviewer’s comments in the revised version of the review. The reference section is now extended from 90 references to 122 references. These references are added in-text wherever necessary. There were a several typos that were fixed. Some paragraphs carrying repetitive messages were combined. Many unclear points have been rephrased to improve the readability of the review. Please find all these changes in red font
Major comment:
- Edit the text to prevent duplication of message. For example, the author often refers to WAVE2 regulating invasion and lamellipodial protrusions, such as line 258, line 272, line 276 This dilutes the message and prevents depth in the review. Line 461 and line 482 have the same message.
Our Response: Line 272 and 276 were deleted due to repetition of the message
Line 482 now line 463 reads as “Despite dramatic advancements in our understanding of WAVEs in cancer cells, it still remains unclear as to which molecule in WAVE signaling pathway could be targeted for an anti-cancer drug therapy”
- Significant improvements must be made to the reference sections. A large fraction of scientific statements is made without listing the appropriate references.
Our Response: We completely agree with the reviewer’s comments. There were many statements without enough references supporting them. The appropriate references are added beyond the reviewer’s comments in the revised version of the review. The reference section is now extended from 90 references to 137 references. These references are added in-text wherever necessary.
- We suggest discussing the molecular basis of the WAVE complex before listing the different type of cancers. Section 3 before section 2.
Our Response: Thank you for noticing this point. We agree that it would be beneficial to understand the molecular basis of WAVE2 before reading its role in different types of cancers. We have now rearranged the text according to the reviewer’s suggestion
Minor comment:
- Verify postal code in affiliation.
Our Response: Postal codes, 44106 for affiliation 1, 44109 for affiliation 2 and 44109 for affiliation 3 have been now added
- Line 17 and 18 are redundant in the abstract.
Our Response: We agree with the reviewer’s comments and the new sentence now reads “Several studies have highlighted the potential for targeting the genes encoding either a part of or a complete protein from WASP/WAVE family as therapeutic strategies for preventing the invasion and metastasis of cancer cells” to avoid redundancy
- Line 37. Is invasion really the initial step in cancer progression? Re-word –
Our Response: We agree to the reviewer’s comment about this sentence. Invasion is not an initial but an early step in cancer progression. The sentence has been reworded (please see line 34)
- Line 46, there seems to be an additional space between shape and Actin.
Our Response: Line 46 is now line 43 and the space has now been deleted
- Line 49, a space is missing between formation and the bracket of the reference.
Our Response: Line 49 is now line 47 and the additional space has been deleted
- Line 54, a space is missing between the comma and 2.
Our Response: Line 54 is now line 52 and the space has been added between the comma and 2
- Line 57, which collectively form a or the VCA region.
Our Response: It should read “the VCA region”. Line 57 is now line 55 and “a VCA region” is replaced by “the VCA region” in the text.
- Line 71, add references regarding the Wiskott-Aldrich syndrome and mention in one sentence what is the syndrome about.
Our Response: Line 71 is now line 76 and the new sentence now reads “The discovery of WASP protein evolved from the study of Wiskott-Aldrich syndrome; it is an X-linked autosomal recessive disorder due to mutation in WAS gene and is characterized by recurrent infection due to immune deficiency microthrombocytopenia, and eczema”
This sentence has now been supported with reference 28
- Line 72 is redundant since the expression pattern of WASP/WAVE is already mentioned line 57 to 61.
Our Response: Line 72 is now line 76. The redundant sentence 72 (about WASP/WAVE expression in blood cells) has been deleted and replaced with
“The discovery of WASP protein evolved from the study of Wiskott-Aldrich syndrome; it is an X-linked autosomal recessive disorder due to mutation in WAS gene and is characterized by recurrent infection due to immune deficiency microthrombocytopenia, and eczema”
This sentence has now been supported with reference 28
- Line 78 add references for the role of WASP in blood and immune response.
Our Response: Line 78 is now line 85-line 89. The new rephrased sentence now reads as “Due to their decreased ability to mount an immune response against foreign invaders, immune problems related to Wiskott-Aldrich syndrome develop in the body. Micro-thrombocytopenia occurs due to the lack of WASP in platelets which impairs their de-velopment leading to reduced size and early cell death [29–33]. Additionally, platelets are cleared from the circulation by the spleen and bone marrow at a faster rate and this could be due to abnormal shape or structure, or the presence of platelet associated immunoglobulins”
References- [28, 34–36] have been added to support this role of WASP in blood and immune response
- Line 80 add references
Our Response: Line 80 is now line 85. References 29,30,31,32 and 33 were added
- Line 85 add references etc…
Our Response: Line 85 is now line 90 and references 28, 34, 35 and 36 were added
- Line 102, downstream of the small Rho GTPase Rac, Line 105 reference, Line 109, complex, which –
Our Response: We agree with the reviewer’s comments and have added the supporting references to the text in line 102, 105 and 109 which is now line 90 to line 110.
The paragraph has been reworded to avoid repetition and the new paragraph now reads “WAVE1, 2 and 3, encoded by WASF1, WASF2 and WASF3 respectively, play a critical role downstream of Rac, a Rho family small GTPase. They are involved in the formation of WAVE regulatory complex (WRC) which in turn regulates the actin cytoskeleton [37]. The WRC is also associated with actin nucleation core Arp2/3 complex while promoting actin polymerization at the leading edge of lamellipodia [38]. WAVE family proteins interact with brain-specific angiogenesis inhibitor 1-associated protein 2 also known as IRSp53 (insulin receptor substrate of 53 kDa) [39, 40]. According to reports the WAVE1 expression is becomes increasingly restricted to CNS over the course of development and homozygous disruption of WAVE1 gene has been shown to result in postnatal lethality. Several other developmental disorders include severe limb weakness, tremors, neuroanatomical malformations etc. [41]. Studies have shown that WAVE2 deficient embryos display growth retardation and some morphological defects such as malformation of ventricles in the developing brain. The WAVE-2 deficient embryonic fibroblasts have shown to exhibit severe growth defects [42]. Studies have shown no abnormal development or function of mouse mammary gland or brain development in the absence of WAVE3 [43]. Since WASP and WAVE proteins are required for several biological processes, such as formation of filopodia and lamellipodia and provide cells with migratory property, they have recently been a topic of great interest in cancer invasion and metastasis. The main focus of this review revolves around WAVE2 in its potential role in the pathogenesis of different types of cancers and it’s signaling events that lead to cancer development, invasion and metastasis.”
- Line 158, that the cisplatin-resistant, remove the “in”
Our Response: Line 158 is now line 181 and the sentence has been grammatically corrected to “Moreover, it has been shown that in the cisplatin-resistant cervical cancer tissues, the expression of SH3BP1, Rac1, and WAVE2 mRNA is significantly upregulated compared to cisplatin sensitive cancer tissues [64].”
- Line 183, according to stats, re-word this sentence.
Our Response: Line 183 is now line 206 and the sentence has been rephrased to “In United States about 1 in 8 men will be diagnosed with prostate cancer during his lifetime. Prostate cancer has accounted for about 13.1% of new cases in men and 10.7% percent of all cancer deaths in 2021”
Reference 63 supports this statement
- Line 196, TNBC is a clinical subtype, not a type of breast cancer.
Our Response: We agree with the reviewer’s suggestions
Line 196 is now line 220 and the sentence has been rephrased to “TNBC is a molecular and clinicopathological subtype of breast cancer that lacks expression of the receptors that are commonly found in this type of cancer (estrogen, progesterone, and HER2), which makes it a more difficult target for treatment”
- Line 209, complex role of WAVE2 complex, reword to remove duplication.
Our Response: Reviewer’s suggestions have been taken into consideration.
Line 209 is now line 233 and the new rephrased sentence now reads “. Overall, their studies present an intricate role of WAVE2 complex in controlling breast epithelial morphogenesis through Abl kinase and the transcription factor Twist1” to avoid duplication of the word “complex”
- Line 284, format of reference is not the same
Our Response: Correction of the reference format (in-text reference number 80) has been made. Please see line 269.
- Figure comments:
- Figure 1. The lamellipodium formation seems to be outside of the cell membrane. Modify the cell membrane do that it reflects that the actine polymerization in inside the cell.
Our Response: We agree with the reviewer’s suggestions. In figure 1, the membrane has been redrawn in a way that it wraps around the actin filaments.
- Figure 1: Also, the consequences of activation on the other component of the WAVE2 complex when they are in the open conformation? Do they maintain their interaction, they should be represented on the figure when activated?
Our Response: We agree with the reviewer’s suggestions and have shown that the other components of WRC maintain their interaction in an open form. Additional interactions between VCA and Arp2/3, Sra and Rac1, proline rich domain and IRSp53 has been now shown in the figure 1 in the active WAVE2 complex

Reviewer 3 Report
The manuscript submitted by Rana et al. provided a comprehensive summary of the current state of knowledge on the topic of molecular signaling mediated by the WAVE and WASP family of proteins as part of the WAVE regulatory complex (WRC) and their role in tumor cell invasion and metastasis.
Novelty:
The novelty of the present article is that it offers an update to the role of specific molecular mechanisms which may allow the development of novel checkpoint inhibitor therapies for various types of malignant tumors. The topic of the article is of particular interest given the high prevalence and mortality of malignant diseases and the ongoing efforts in discovering novel therapeutic strategies capable of combating tumor invasion and metastasis. The authors provide a synthesis of the components underlying the complex immunological mechanisms mediated by the WAVE/WASP family of proteins which have been investigated so far.
The present review is balanced and the premise of the article is sensible. The authors have adequately interpreted and presented the data currently available in the literature, in addition to presenting competing hypotheses in a comprehensive manner and discussing the shortcomings of the proposed ideas. The majority of the references included in the article consist of primary research conducted over the past 10 years.
Structure:
The structure of the article is accessible and the abbreviations and acronyms are useful and standard. The take-home messages and the justification for writing the review are clear. The manuscript's figures complement the reviewed information nicely. A minor issue might be the fact that the figures have several abbreviations which are not explained in the description. Although these items are discussed in the text, perhaps a description of the abbreviations would improve the readers' understanding of the subject.
Contents:
The title and abstract are appropriate for the content of the text. However, the keywords which were selected are not particularly appropriate for the contents of the article. The terms "breast cancer" and "triple negative breast cancer", although discussed in the text, do not constitute the main focus of the paper, since other types of tumors are investigated in a similar manner (colorectal, prostate, pancreatic and cervical cancer).
In line 81-83, the authors state that: "microthrombocytopenia develops in individuals with Wiskott-Aldrich syndrome due to impaired process of removing platelets from circulation and taking them to spleen for the destruction". A revision of this sentence would improve the flow and readability of the text.
Line 92: " is becomes"- requires revision. In addition, a number of statements are formulated in this paragraph which are not supported by appropriate citations.
Line 139: " Colorectal cancer is the second most common cause of cancer death in the United states and according to the American cancer society, the lifetime risk of developing the colorectal cancer is about 4.3% for men and 4% for women" - this statement is not entirely accurate since the incidence and mortality associated with CRC is different for men and women. In addition, the authors do not provide adequate citations supporting this claim.
Line 150: "According to WHO Cervical cancer is the fourth common cancer worldwide affecting women in cancer incidence and mortality rates (include recent statistics)" - This statement should be revised due to the fact that the meaning of the sentence is unclear.
Line 152: " It has accounted for about 0.8 percent of all new cancer diagnoses and 0.7 percent of all cancer deaths for 2021" - the authors do not mention the appropriate citation for supporting this claim.
Line 153: "cervical cancer is most commonly diagnosed in women between the age group of 35-44 years with an average age of diagnosis of 50 years" - this statement should be revised as it presents contradictory information. It is unclear how the average age of diagnosis is situated outside the specified interval for the most common age of diagnosis.
Line 180: " Prostate cancer is one of the most common cancer among men and is the second leading cause of cancer death in American men" - this statement is unsupported by appropriate citations. The statement that it is frequent among the male population is redundant.
Line 183 " According to stats" - requires revision due to the colloquial character of the phrase, which is somewhat inappropriate for a scientific publication
Line 250 - "growth factors" - is mentioned twice, possibly due to a redacting error
Line 432-433 - "regulates the progression of cancer (colorectal liver) by regulating TGF-B" - the statement included in the parentheses requires revision in order to improve the readers' understanding of the discussed topic
Line 491: " necessary to determine exactly what upstream or downstream pathway acts as a switch do drive cells into various modes of cancer invasion" - statement requires revision due to a possible redacting error
Lastly, while the use of language is mostly sound, there are a few instances where certain points are unclear, making the narrative difficult to follow. A revision of spelling and syntax is required in order to improve the flow and readability of the text.
Author Response
Subject: Revision and resubmission of manuscript number: and response to the reviewers
Dear Reviewer,
We would like to thank you for extending us the opportunity to submit a revised draft “ROLE WAVE2 SIGNALING IN CANCER” for publication in Biomedicines Journal. We appreciate your time and efforts to review and improve the article and providing the necessary feedback. We have incorporated most of the suggestions made by the reviewers. These changes are marked in red text within the manuscript. Please see below for a point-by-point response to the reviewer.
Reviewer 3:
Comments and Suggestions for Authors
The manuscript submitted by Rana et al. provided a comprehensive summary of the current state of knowledge on the topic of molecular signaling mediated by the WAVE and WASP family of proteins as part of the WAVE regulatory complex (WRC) and their role in tumor cell invasion and metastasis.
Novelty:
The novelty of the present article is that it offers an update to the role of specific molecular mechanisms which may allow the development of novel checkpoint inhibitor therapies for various types of malignant tumors. The topic of the article is of particular interest given the high prevalence and mortality of malignant diseases and the ongoing efforts in discovering novel therapeutic strategies capable of combating tumor invasion and metastasis. The authors provide a synthesis of the components underlying the complex immunological mechanisms mediated by the WAVE/WASP family of proteins which have been investigated so far.
The present review is balanced and the premise of the article is sensible. The authors have adequately interpreted and presented the data currently available in the literature, in addition to presenting competing hypotheses in a comprehensive manner and discussing the shortcomings of the proposed ideas. The majority of the references included in the article consist of primary research conducted over the past 10 years.
Structure:
The structure of the article is accessible and the abbreviations and acronyms are useful and standard. The take-home messages and the justification for writing the review are clear. The manuscript's figures complement the reviewed information nicely.
- A minor issue might be the fact that the figures have several abbreviations which are not explained in the description. Although these items are discussed in the text, perhaps a description of the abbreviations would improve the readers' understanding of the subject.
Our Response: We agree with the reviewer’s comments and the abbreviations have now been described in the figure legends to improve understanding of the figures
Contents:
- The title and abstract are appropriate for the content of the text. However, the keywords which were selected are not particularly appropriate for the contents of the article. The terms "breast cancer" and "triple negative breast cancer", although discussed in the text, do not constitute the main focus of the paper, since other types of tumors are investigated in a similar manner (colorectal, prostate, pancreatic and cervical cancer).
Our Response: We agree with the reviewer’s suggestions. The keywords “breast cancer” and “Triple negative breast cancer” have now been deleted from the keywords section
- In line 81-83, the authors state that: "microthrombocytopenia develops in individuals with Wiskott-Aldrich syndrome due to impaired process of removing platelets from circulation and taking them to spleen for the destruction". A revision of this sentence would improve the flow and readability of the text.
Our Response: We have taken the reviewer’s suggestions into consideration and the Line 81- 83 is now line 85-89. The new rephrased sentence now reads as “Microthrombocytopenia occurs due to the lack of WASP in platelets which impairs their development leading to reduced size and early cell death [29–33]. Additionally, platelets are cleared from the circulation by the spleen and bone marrow at a faster rate and this could be due to abnormal shape or structure, or the presence of platelet associated immunoglobulins [28, 34–36]”
- Line 92: " is becomes"- requires revision. In addition, a number of statements are formulated in this paragraph which are not supported by appropriate citations.
Our Response: We agree with the reviewer and multiple paragraphs are now merged due to repetition of the message. Appropriate references are now added. Please find the updated paragraph and the supporting references in line 90-110.
The new statement now reads “WAVE1, 2 and 3, encoded by WASF1, WASF2 and WASF3 respectively, play a critical role downstream of Rac, a Rho family small GTPase. They are involved in the formation of WAVE regulatory complex (WRC) which in turn regulates the actin cytoskeleton [37]. The WRC is also associated with actin nucleation core Arp2/3 complex while promoting actin polymerization at the leading edge of lamellipodia [38]. WAVE family proteins interact with brain-specific angiogenesis inhibitor 1-associated protein 2 also known as IRSp53 (insulin receptor substrate of 53 kDa) [39, 40]. According to reports the WAVE1 expression is becomes increasingly restricted to CNS over the course of development and homozygous disruption of WAVE1 gene has been shown to result in postnatal lethality. Several other developmental disorders include severe limb weakness, tremors, neuroanatomical malformations etc. [41]. Studies have shown that WAVE2 deficient embryos display growth retardation and some morphological defects such as malformation of ventricles in the developing brain. The WAVE-2 deficient embryonic fibroblasts have shown to exhibit severe growth defects [42]. Studies have shown no abnormal development or function of mouse mammary gland or brain development in the absence of WAVE3 [43]. Since WASP and WAVE proteins are re-quired for several biological processes, such as formation of filopodia and lamellipodia and provide cells with migratory property, they have recently been a topic of great interest in cancer invasion and metastasis. The main focus of this review revolves around WAVE2 in its potential role in the pathogenesis of different types of cancers and it’s signaling events that lead to cancer development, invasion and metastasis.”
- Line 139: " Colorectal cancer is the second most common cause of cancer death in the United states and according to the American cancer society, the lifetime risk of developing the colorectal cancer is about 4.3% for men and 4% for women" - this statement is not entirely accurate since the incidence and mortality associated with CRC is different for men and women. In addition, the authors do not provide adequate citations supporting this claim.
Our Response: Line 139 is now line 162. The cancer statistics were verified and calculated for both sexes combined. The new sentence now reads as “Colorectal cancer is the third most common cause of cancer death in both men and women in the United States. It has accounted for about 7.88 percent of 2021 cancer diagnoses in both men and women and 8.7 percent of all cancer deaths in the year 2021”
Reference 58 supports the above statistics on colorectal cancer
- Line 150: "According to WHO Cervical cancer is the fourth common cancer worldwide affecting women in cancer incidence and mortality rates (include recent statistics)" - This statement should be revised due to the fact that the meaning of the sentence is unclear.
Our Response: We agree with the reviewer’s comments. Line 150 is now line 174. The unclear statement is corrected for the most recent statistics from reference number 58.
The new paragraph now reads as “According to most recent cancer statistics, Cervical cancer is the fourth common cancer in United States”.
- Line 152: " It has accounted for about 0.8 percent of all new cancer diagnoses and 0.7 percent of all cancer deaths for 2021" - the authors do not mention the appropriate citation for supporting this claim.
Our Response: Line 152 is now line 175. The correct numbers are updated according to the 2021 statistics and the new sentence reads as “It has accounted for about 1.5 percent of 2021 cancer diagnoses in women and 1.48 percent of all cancer deaths (in women) in the year 2021 [58]”
Reference 58 supports the above information
- Line 153: "cervical cancer is most commonly diagnosed in women between the age group of 35-44 years with an average age of diagnosis of 50 years" - this statement should be revised as it presents contradictory information. It is unclear how the average age of diagnosis is situated outside the specified interval for the most common age of diagnosis.
Our Response: We agree with the reviewer’s comments. Line 153 is now line 176 and the contradictory information was removed from the text
- Line 180: " Prostate cancer is one of the most common cancer among men and is the second leading cause of cancer death in American men" - this statement is unsupported by appropriate citations. The statement that it is frequent among the male population is redundant.
Our Response: We agree with the reviewer’s comments.
The statement that it is frequent among male population has been deleted due to the redundancy.
Line 180 is now line 204. Most recent statistics according to the year 2021 have been added.
The new sentence now reads “According to 2021 statistics, prostate cancer is first leading cancer among men and is the second leading cause of cancer death, behind only lung cancer in the United States [58]. In United States about 1 in 8 men will be diagnosed with prostate cancer during his lifetime. Prostate cancer has accounted for about 13.1% of new cases in men and 10.7% percent of all cancer deaths in 2021 [58].”
Reference 58 support the above statistics
- Line 183 " According to stats" - requires revision due to the colloquial character of the phrase, which is somewhat inappropriate for a scientific publication
Our Response: We agree with the reviewer’s comments
Line 183 is now line 206 which has been rephrased “In United States about 1 in 8 men will be diagnosed with prostate cancer during his lifetime”
The inappropriate sentence was deleted
- Line 250 - "growth factors" - is mentioned twice, possibly due to a redacting error
Our Response: We agree with the reviewer’s comments.
Line 250 is now line 136. The paragraph on the WAVE regulatory complex was moved before the role of WAVE2 in cancer because it would be beneficial to understand the molecular basis of WAVE2 before reading its role in different types of cancers.
The redacting error on “Growth factors” was taken care of. Please see line 135 and 136 for the correction
- Line 432-433 - "regulates the progression of cancer (colorectal liver) by regulating TGF-B" - the statement included in the parentheses requires revision in order to improve the readers' understanding of the discussed topic
Our Response: We agree with the reviewer’s comments. The statement has been revised in line 417 which now reads as “In colon cancer liver metastasis (CLM) patients, WAVE2 expression was strongly correlated to microvessel density in hepatic metastasis. Moreover, WAVE2 promotes the progression of CLM by regulating TGF-B and Hippo pathways via effector yes as-sociated protein (YAP1) in the hepatic stellate cells [62]”
- Line 491: " necessary to determine exactly what upstream or downstream pathway acts as a switch do drive cells into various modes of cancer invasion" - statement requires revision due to a possible redacting error
Our Response: We agree with the reviewer’s comments and the error has been corrected. Line 491 is now line 479 which reads “However, to further validate WAVE2 as an anticancer target, it is necessary to deter-mine exactly how upstream and downstream pathway interplay to drive cells into various modes of cancer invasion”
- Lastly, while the use of language is mostly sound, there are a few instances where certain points are unclear, making the narrative difficult to follow. A revision of spelling and syntax is required to improve the flow and readability of the text.
Our Response: Thank you for suggesting this. We agree with the reviewer’s suggestions. There were a several typos that were fixed. Some paragraphs carrying repetitive messages were combined. Many unclear points have been rephrased to improve the readability of the review. Please find all these changes in red font

Round 2
Reviewer 2 Report
I recommend to accept this review and I am satisfied by the changes the authors made to improve their manuscript